# Effects of Perennial Alfalfa on the Structure and Function of Soil Micro-Food Webs in the Loess Plateau

**DOI:** 10.3390/microorganisms12112268

**Published:** 2024-11-08

**Authors:** Liangliang Li, Jianxia Tian, Zhuzhu Luo, Lingling Li, Yining Niu, Fasih Ullah Haider, Lili Nian, Yaoquan Zhang, Renyuan He, Jiahe Liu

**Affiliations:** 1Grassland Science College, Gansu Agricultural University, Lanzhou 730070, China; 18394797671@163.com; 2College of Resources and Environmental Sciences, Gansu Agricultural University, Lanzhou 730070, China; 17899315620@163.com (J.T.); liujiahe@gsau.edu.cn (J.L.); 3State Key Laboratory of Aridland Crop Science, Gansu Agricultural University, Lanzhou 730070, China; lill@gsau.edu.cn (L.L.); niuyn@gsau.edu.cn (Y.N.); 4Key Laboratory of Vegetation Restoration and Management of Degraded Ecosystems, South China Botanical Garden, Chinese Academy of Sciences, Guangzhou 510650, China; fasihullahhaider281@gmail.com; 5College of Forestry, Gansu Agricultural University, Lanzhou 730070, China; nll18893814845@163.com (L.N.); zhangyqgs@163.com (Y.Z.); 18393414923@163.com (R.H.)

**Keywords:** planting years, metabolic footprint, soil micro-food web, correlation network

## Abstract

The Loess Plateau is one of the most vulnerable areas in the world. Numerous studies have been conducted to investigate alfalfa fields with different planting years. Soil microorganisms and nematodes are vital in ecosystem functionality and nutrient cycling. Therefore, comprehending their response to alfalfa fields with varying years of planting is essential for predicting the direction and trajectory of degradation. Alfalfa fields with different planting years (2 years, 9 years, and 18 years) were used as the research object, and farmland was used as the control (CK). High-throughput sequencing and morphological methods determined the community composition of microorganisms and nematodes. Carbon metabolic footprints, correlation networks, and structural equations were used to study soil microorganisms and nematode interactions. Principal component analysis (PCA) results showed that alfalfa fields with different planting years significantly impacted soil microorganisms and nematode community structures. Planting alfalfa significantly increased the nematode channel ratio (NCR) and Wasilewska index (WI), but significantly reduced the soil nematode PPI/MI and dominance (λ). The correlation network results indicated that, for the 2-year and 18-year treatments, the total number of links and positive links are higher than other treatments. Conversely, the 9-year treatment had fewer positive links and more negative links compared to other treatments. Additionally, the keystone species within each network varied based on the treatment years. Structural equation results show that alfalfa planting years directly impact soil fungal community structure and plant-parasitic nematodes’ carbon metabolism omnivorous-predatory nematodes. Furthermore, the carbon metabolism of omnivorous-predatory nematodes directly influences soil organic carbon fixation. Moreover, as the duration of alfalfa planting increases, the metabolic footprint of plant-parasitic nematodes decreases while that of omnivorous-predatory nematodes rises. Among treatments varying in alfalfa planting durations, the 9-year treatment exhibited the most incredible energy conversion and utilization efficiency within the soil food web, demonstrating the most stable structure. This study reveals optimal alfalfa planting duration for soil ecosystem stability in the Loess Plateau. Future research should explore sustainable crop rotations and alfalfa–soil–climate interactions for improved agricultural management.

## 1. Introduction

The most significant organic carbon pool in terrestrial ecosystems is soil, which stores more carbon than the combined vegetation and atmospheric pools [1]. Soil organic carbon serves a crucial function by contributing to ecosystem services and exerting regulatory control over the global carbon cycle. Even minor global soil organic carbon pool fluctuations can profoundly influence atmospheric CO_2_ concentrations [2]. Changes in soil organic carbon depend on the balance between plant input and decomposition output of microorganisms and nematodes, and these processes will be completed in the soil micro-food web [3]. A complex food chain relationship exists between soil nematodes and microorganisms, forming a soil micro-food web [1]. The soil micro-food web supports ecological functions, encompassing everything from the decomposition and release of soil nutrients to their uptake by plants. Soil nematodes and other micro-food web members play critical roles in these processes. Their significance has become increasingly apparent with advancements in subterranean ecology and improvements in research methodologies [2]. Several studies have demonstrated that when soil nematodes feed on microorganisms, they excrete excess nutrients beyond their metabolic requirements, thereby enriching the effective nutrient content in the soil [3]. Concurrently, nematode predation promotes microbial reproduction and metabolic activities, increasing soil respiration and enzyme activity [4]. Given that the predatory interactions between nematodes and microorganisms primarily form the soil micro-food web, organisms within the soil micro-food web may have a more prominent influence on the mineralization and release of soil nutrients than soil microorganisms alone, thereby enhancing the effective nutrient content [2,5]. Furthermore, certain studies suggest that soil nematodes play a pivotal role in balancing nutrient competition between plants and microorganisms [6]. When nutrient competition arises between soil microorganisms and plants [7], nematode predation on microorganisms can enhance the competitive advantage of plants [8]. Nematodes at higher trophic levels, notably omnivorous and predatory nematodes, prey on nearly all smaller organisms, including plant-parasitic nematodes detrimental to plants [9]. This predation will also positively affect plant growth (Figure 1) [10].

Soil nematodes comprise various types, such as bacterivorous, fungivorous, plant-parasitic, and omnivorous-predatory nematodes. They are categorized into three different carbon flow channels: through bacteria, fungi, and root channels (Figure 1). Increasing evidence suggests that the interactions among these pathways significantly influence the regulation of soil food webs [11,12,13]. To provide a more precise depiction of the soil food web channels and carbon dynamics in nematode production and respiration, Ferris [11] introduced the concept of the soil nematode metabolic footprint (NMF), which reflects the carbon amount of soil nematodes in the food web. Metabolic footprints can be classified into enrichment and structural categories. The enrichment footprint represents the metabolic response of nematodes to rapid environmental resource enrichment. In contrast, the structural metabolic footprint relates to high-trophic-level nematodes that possess regulatory roles within the food web [14]. The introduction of this concept has advanced our comprehension of carbon flows in soil food webs. Consequently, the potential of soil nematodes as biological indicators, reflecting the impacts of human activities and soil disturbances on soil food webs, has attracted heightened scholarly interest [15]. Recent research has focused on how the metabolic footprint of soil nematodes during vegetation restoration affects soil carbon flux. However, an agreement on the evolving patterns and underlying drivers of soil carbon flux during vegetation restoration has yet to be reached.

The Loess Plateau in northwest China has a long history of farming. Due to the unique ecological environment and soil parent material, the Loess Plateau suffers from severe water and soil erosion, resulting in one of the world’s most vulnerable locations [16,17,18]. Large-scale vegetation restoration and reconstruction projects, such as converting farmland to forest and grassland, have significantly increased vegetation coverage on the Loess Plateau [19]. Vegetation restoration enhances organic matter input into the soil, while microbial activity contributes to the decomposition and transformation of organic carbon, ultimately affecting soil organic carbon storage [20]. On the Loess Plateau, vegetation restoration supports carbon fixation and improves biodiversity, ecosystem stability, and ecosystem services—all vital components of global carbon sequestration [21]. As a critical ecological region in China, the enhanced carbon sink function of the Loess Plateau positively influences both regional and global carbon cycles. These roles highlight the significance of the Loess Plateau in global carbon management. Alfalfa (*Medicago sativa*) is the favoured forage for converting farmland to forest and grassland within the ecologically sensitive regions of the Loess Plateau. It has ecological solid adaptability, but the rhizobia in the roots increase soil organic matter through nitrogen fixation [22] and improve the soil aggregate structure [23]. In addition, alfalfa is a high-protein forage crop, and its cultivation provides high-quality feed resources that support the development of animal husbandry [24]. Feeding livestock with alfalfa silage or hay can significantly improve meat and milk quality while reducing reliance on costly fertilizers [25]. Consequently, alfalfa is pivotal in enhancing the surrounding biological environment and advancing animal husbandry methods. Characterized by its axial-root structure, alfalfa boasts a robust root system. However, prolonged cultivation may result in continuous cropping failure, depleting soil moisture and phosphorus. Such circumstances can subsequently reduce grass yield, deteriorate soil quality, and potentially induce autotoxic effects that reduce the quality of succeeding crops [26]. In recent years, as research on the interactions between the above-ground and underground components of plants and soil organisms has advanced, the complex relationship between soil organisms and various soil ecological processes—including plant growth, aggregate formation, soil respiration, nutrient cycling, organic matter degradation, and carbon storage—has gained more apparent recognition [27,28]. Therefore, selecting a suitable planting year benefits the growth of alfalfa and improves soil fertility. Despite the importance of alfalfa in ecological restoration on the Loess Plateau, there is limited understanding of how different planting durations affect soil micro-food web dynamics and decomposition pathways. Therefore, the current study aimed to address this knowledge gap by investigating the effects of various alfalfa planting years on soil microbial and nematode communities and their interactions within the soil micro-food web. Using an 18-year alfalfa planting experiment, the objectives of this study were to:Assess the impact of different planting durations on soil microbial and nematode community structures.Elucidate the influence of planting years on decomposition pathways within the soil micro-food web.Identify optimal planting durations for enhancing soil fertility and ecosystem stability.

It was hypothesized that different planting years would significantly influence microbial and nematode communities, affecting decomposition pathways and soil carbon dynamics. It was also anticipated that specific planting durations would activate and regulate distinct channels and mechanisms in the decomposition pathways, potentially leading to variations in soil organic carbon storage and ecosystem functionality. This study will provide insights into the long-term effects of alfalfa cultivation on soil ecology, offering valuable guidance for sustainable land management practices on the Loess Plateau and similar vulnerable ecosystems.

## 2. Material and Methods

### 2.1. Site Description

The study was conducted at an experimental site at the Comprehensive Experimental Station for Dry Farming of the Loess Plateau, affiliated with Gansu Agricultural University, in northwestern China. The site is in Dingxi City, Gansu Province, at 35°28′ N latitude and 104°44′ E longitude, with an average elevation of 1970 m. The region receives an average of 2476.6 h of annual sunshine, with solar radiation reaching up to 592.9 kJ·cm^−2^ per unit area. The frost-free period lasts approximately 140 days. The average yearly temperature is around 6.4 °C, with fluctuations ranging from 5.8 °C to 6.8 °C. The mean annual precipitation is 390 mm, while annual evaporation averages 1531 mm. The precipitation guaranteed at 80% reliability is 365 mm. The yearly evaporation is 1531 mm, and the mean annual rainfall is 390 mm. The test area is in a semi-arid zone in the mid-temperate zone, a typical dry farming area with one crop per year. The soil is loess, with loose soil, deep soil layer, uniform texture.

### 2.2. Experimental Design and Soil Sampling

The experiment was conducted on alfalfa grassland, which was converted from farmland and sown in 2003 (18 years), 2012 (9 years), and 2019 (2 years). Concurrently, local farmland, where the primary crop was corn, was selected as the control (CK). Each treatment plot had three replicates, and each plot measured 3 m × 7 m. The plots were organized in a randomized block design. The alfalfa variety was “Longdong Alfalfa”, with a seeding rate of 18 kg·hm^−2^ and a wide row spacing of 20 cm. When alfalfa was planted (in 2003, 2012, and 2019), 105 kg·hm^−2^ of pure nitrogen and pure P_2_O_5_ seed fertilizers were spread. Subsequently, no fertilizer or irrigation was applied during the entire growth period of the alfalfa grassland. Harvesting took place once in June and again in October every year. The corn variety “Xianyu 335” was used in the CK group, with a planting density of 52,500 plants·hm^−2^. Annually, before sowing and following the local traditional fertilization method, 200 kg·hm^−2^ pure nitrogen and 105 kg·hm^−2^ P_2_O_5_ were manually applied. No additional fertilizers were used during the corn growth period, and post-harvest, neither the corn plants nor the roots were returned to the field. These specific treatment years were selected to study the life cycle dynamics of alfalfa, a perennial plant with distinct growth stages. Alfalfa fields at 2, 9, and 18 years were chosen as treatment groups, representing different stages in the alfalfa growth cycle, from young to mature. By comparing these varying ages of alfalfa fields, we can observe changes in soil microorganisms, nematodes, and soil properties as the cultivation years increase. Long-term observation and data collection will provide a better understanding of the potential of alfalfa cultivation in semi-arid regions and its impact on the environment.

In June 2021, during the blooming stage of the first alfalfa crop and the jointing stage of corn, soil samples were collected from a 0–20 cm depth around the root perimeter using the 5-point sampling method. After removing debris, such as stones, gravel, and plant residues, the samples were combined into a single composite sample. After removing materials like stones, gravel, and plant remains, the samples were blended into a single composite sample. Each soil sample was then categorized into four portions. The first portion was placed in sterilized centrifuge tubes and stored in a foam box with ice packs for transportation to the laboratory, where they were then kept at −80 °C. The second portion was stored in sterilized ziplock bags in a foam box with ice packs, refrigerated at 4 °C until its subsequent use for nematode extraction and identification. Additional samples were put in specific aluminum boxes to measure the soil moisture content of the experimental region. The remaining samples were air-dried in a cool, well-ventilated location at the laboratory, after which their physicochemical properties were determined. These properties include soil pH, organic matter content, total nitrogen content, particle composition, etc., essential for evaluating soil quality and formulating reasonable agricultural management measures. Through such sampling and processing methods, researchers can obtain detailed data on soil microbial diversity, nematode community structure, soil moisture status, and soil physical and chemical properties to better understand the function of soil ecosystems and the impact of alfalfa planting on soil.

### 2.3. Analysis of Soil Physicochemical Properties

Fresh soil samples were uniformly dried at 105 °C to establish a constant weight for determining soil moisture content (SM). The pH of the soil was measured using a pH meter with a soil-to-water ratio of 1:2.5 extract. Soil organic carbon was calculated using the dichromate oxidation method [29]. Soil total nitrogen (TN) was determined using the Kjeldahl method [30]. The molybdenum blue method assessed soil total phosphorus (TP) [31]. The Olsen method determined soil-available phosphorus (AP) [32]. Analysis of nitrate nitrogen (NO_3_^−^-N) and ammonium nitrogen (NH_4_+-N) was conducted using a continuous flow analyzer [33].

### 2.4. DNA Extraction and High-Throughput Sequencing

Soil total genomic DNA was extracted according to the Power Soil^®^ DNA Kit (Omega Bio-tek, Norcross, GA, USA). The extracted DNA underwent assessment for purity via 1.0% agarose gel electrophoresis [34], accompanied by ethidium bromide staining, followed by detection using a gel imaging system. The V4-V5 region of the 16S rRNA gene of bacteria was amplified with the primers 515F (5′-GTGCCAGCMGCCGCGG-3′) [35] and 907R (5′-CCGTCAATTCMTTTRAGTTT-3′) [36]. For fungi, the ITS1 region was amplified using the primer pair ITS1F(5′-CTTGGTCATTTAGAGGAAGTAA-3′) and ITS1R (5′-GCTGCGTTCTTCATCGATGC-3′) [36]. The PCR procedure utilized Trans Gen AP221-02 Trans StartFastpfu DNA Polymerase (TIANGEN, Beijing, China) with an ABI Gene Amp^®^ 9700 instrument (ABI, Foster City, CA, USA). PCR products from the same sample were combined and analyzed through 2% agarose gel electrophoresis. Subsequently, the AxyPrep DNA Gel Recovery Kit (AXYGEN, Union City, CA, USA) was employed for gel excision and PCR product retrieval, which were then eluted with Tris HCl and again evaluated via 2% agarose electrophoresis. For quantification, the PCR products were subjected to the QuantiFluor™-ST Blue Fluorescence Quantification System (Promega Corporation, Madison, WI, USA) [37,38]. After quantification and homogenization, the PCR products were consolidated, Miseq libraries were constructed, and sequencing was conducted using the high-throughput sequencing platform offered by Shanghai Meiji Biomedical Technology Co. (Shanghai, China).

### 2.5. Analysis of Soil Nematode Communities

The shallow dish method was used to extract soil nematodes. A sample of 100.0 g of fresh soil was weighed and placed on filter paper within a shallow dish. Slowly pour water into the shallow dish until a thin water film forms on the soil surface, incubating the soil for 48 h at room temperature. Nematodes were subsequently separated using a 500-mesh sieve (pore size 25 μm). Nematodes were collected, killed in a 60 °C water bath, and then preserved by being fixed in a 4% formalin solution. Nematodes were counted under a microscope and then converted into the number of nematodes per 100 g of dry soil [39]. The classification and identification of soil nematodes are based on the webpage established by Ferris at http://Nemaplex.ucdavis.edu, accessed on 1 December 2022. Genus-level identification was performed microscopically. Based on their eating habits and head morphology, nematodes were further divided into four trophic groups: plant parasites (PP), omnivorous predators (OP), bacterial predators (BF), and fungal predators (FF) [40].

### 2.6. Ecological Function Index Calculation

The soil nematode Shannon diversity index (H), dominance index (λ), nematode pathway ratio (NCR), Wasilewska index (WI), plant-parasitic nematode maturity index, and free-living nematode maturity index ratio (PPI/MI) were computed [41,42]. Additionally, the soil microbial Chao1 index [43] and Shannon diversity index (H) [44] were determined.
(1)$$H=-\sum_{i=1}^{s}pi*lnpi$$
(2)$$\lambda =\sum {pi}^{2}$$
(3)$$MI=\sum v_{i}f_{i}$$
(4)$$PPI=\sum v_{i}f_{i}^{\prime }$$
(5)$$NCR=\frac{BF}{BF+FF}$$
(6)$$WI=\frac{BF+FF}{pp}$$
where pi represented the relative abundance of taxon i in the sample, S denoted the total number of nematode genera in the community, pi′ referred to the proportion of nematode abundance in taxon i within its trophic group relative to the total nematode abundance, $v_{i}$ indicated the cp value for taxon i, and $f_{i}$ represented the proportion of free-living nematodes in taxon i. Additionally, $f_{i}^{\prime }$ denoted the proportion of plant-parasitic nematodes in taxon i, BF, FF, and PP referred to the abundance of bacterivorous, fungivorous, and plant-parasitic nematodes, respectively.

### 2.7. Metabolic Footprinting and Faunal Analysis of Nematodes

The enrichment index (EI) was calculated as EI = 100 × (e/((e + b))) and the structural index (SI) as SI = 100 × (s/((s + b))), where, e represented the enrichment component (BF1 and FF2), b represented the basal food web component (BF2 and FF2), and s represented the structural component (BF3–BF5, FF3–FF5, OP3–OP5, and PP2–PP5). Here, BF, FF, OP, and PP refer to the trophic groups: bacterivorous, fungivorous, omnivorous-predatory, and plant-parasitic nematodes, respectively, with the associated numbers indicating specific cp values.

The nematode biomass (fresh weight, W) mentioned in the “Nematode Information System” (https://nematode.unl.edu/index.html, accessed on 1 December 2022) was used to calculate the nematode metabolic footprint (NMF).
$$NMF=\Sigma (N_{t}\times (0.1\times (W_{t}÷m_{t})+0.273(W_{t}^{0.75})))$$
where $N_{t}$ refers to the abundance of t-type nematode group; $m_{t}$ refers to the c-p value of t-type nematode group; and $W_{t}$ refers to the biomass of t-type nematode group [11].

Based on the enrichment footprint (Fe) and structural footprint (Fs), four coordinate points were determined in quadrants A, B, C, and D, centered around the (SI, EI) coordinate. These points are (SI, EI − 0.5Fe/k), (SI + 0.5Fs/k, EI), (SI, EI + 0.5Fe/k), and (SI − 0.5Fs/k, EI), where “k” represents the transformation constant. The functional metabolic footprint (NMF) of the nematode is the area of the quadrilateral circled by connecting the four coordinate points sequentially.

### 2.8. Soil Food Web Carbon Flow Analyses

The energy flow analysis of the soil food web refers to the method of Ferris et al. [44]. The triangle’s lower left corner’s coordinate point is (0, 0), the lower right corner’s coordinate point is (100, 0), and the vertex’s coordinate point is (50, 86.6). Geometry knowledge calculated the coordinate points determining the triangle’s center point and the three sides’ midpoints. Seven coordinate points of each treatment were calculated from the relative metabolic footprints of bacterivorous, fungivorous, and plant-parasitic nematodes. After setting the coordinate points of each process, we selected all of them to draw a scatter diagram, in which the drawing of the three sides of the triangle and the midline of each side can be presented by adding a solid line or a dashed line, finally deleting the coordinate axis.

### 2.9. Statistical Analysis

One-way analysis of variance (ANOVA) was used to assess whether different alfalfa planting years significantly affected soil microbial communities, soil nematode communities, and soil physicochemical properties. For significant results, the least significant difference (LSD) test was applied to compare the mean values of all soil parameters (*p* < 0.05). Data, charts, and graphs for soil properties, soil microbial communities, and soil nematode communities were generated using SPSS Statistics 22 (SPSS Inc., Chicago, IL, USA) and Origin 2021. Canoco 5.0 software was used to identify the main environmental factors affecting soil microorganisms and nematodes and determine correlations between microbial and nematode genera and environmental factors. Correlation ecological network analysis was conducted using the “igraph” package in R1.6.0, primarily to evaluate how interactions among bacteria, fungi, and nematodes influence soil ecosystem health and function. This analysis also represented different relationship types, such as symbiosis, predation, or competition. Structural equation model analysis was performed with the “piecewise SEM” package in R2.1.2, examining both the direct and indirect effects of planting years on bacteria, fungi, and various nematode groups (bacterivorous, fungivorous, plant-parasitic, and predatory/omnivorous nematodes) as well as the effects of the food web comprising bacteria, fungi, and nematodes on carbon storage.

## 3. Results

### 3.1. Environmental Factors

As the alfalfa planting years increase (Appendix A), there is a decline in soil moisture content, and the alfalfa field was significantly lower than that of farmland soil (CK) (*p* < 0.05). The soil organic carbon content increases, with the content in the 18-year alfalfa field being significantly higher than other treatments (*p* < 0.05). Total nitrogen content also observed a significant rise (*p* < 0.05). Total phosphorus content showed a trend of first growing and then dropping, with the 18-year alfalfa field having significantly lower content than the 9-year alfalfa field, and the alfalfa field was considerably lower than CK (*p* < 0.05). The available phosphorus content decreased, with the 2-year alfalfa field showing significantly higher values than the other alfalfa growth years (*p* < 0.05), and the content in the alfalfa field is considerably less than in farmland soil (*p* < 0.05). All treatments do not significantly differ from one another in terms of pH or soil bulk weight.

### 3.2. Composition and Differences of Microbial and Nematode Communities

Lefse (LDA effect size) analysis of microorganisms and nematodes was performed on alfalfa fields with planting years of 2 years, 9 years, and 18 years and CK soil samples (Figure 2). At the soil nematode genus level, 13 genera—such as *Meloidogyne*, *Hoplolaimus*, *Acrobeles*, and *Campydora*—showed significant differences among the different alfalfa planting years (*p* < 0.05). At the bacterial genus level, six genera—including *Nocardioides*, *MND1*, *Pirellula*, and *Pir4_lineage*—demonstrated significant differences (*p* < 0.05). Meanwhile, at the fungal genus level, 18 genera like *Leohumicola*, *Ascochyta*, *Ramophialophora*, and *Lectera* displayed substantial differences among the varying alfalfa planting years (*p* < 0.05).

### 3.3. Principal Component Analysis of Soil Microorganisms and Nematodes

Principal coordinate analysis (PCA) based on the Bray-Curtis algorithm determined the effects of different years of alfalfa fields on the structure of microbial and nematode communities (Figure 3). The PCA divided the bacterial, fungal, and nematode communities into three distinct groups: CK, a combined group of 9-year and 18-year alfalfa fields, and a 2-year alfalfa field group. There were significant changes in the community structures of soil fungi, bacteria, and nematodes across alfalfa fields with different planting years. Specifically, the community structure in the 2-year alfalfa field was far away from that of other treatment groups, indicating that the community structure differed most from that of different treatments. Further, ANOSIM tests showed significant differences in soil microbial and nematode communities across planting years (*p* < 0.05).

### 3.4. Ecological Function Index of Soil Microorganisms and Nematode Communities

The soil nematode Shannon diversity index decreased and then increased, with the value for the 18-year alfalfa field significantly surpassing those of the 2-year and 9-year fields (*p* < 0.05). The NCR and WI indices for soil nematodes exhibited a significant increasing trend (*p* < 0.05) and were distinctly higher in the 18-year alfalfa field than in other planting years. Conversely, the PPI/MI and dominance index (J) for soil nematodes showed a significant decreasing trend (*p* < 0.05) and were substantially lower in the 18-year field relative to other treatments. The soil bacterial Shannon diversity index and Chao1 index were not significantly different. There was no significant difference in the Shannon diversity index of soil fungi. In contrast, the Chao1 index first increased and then decreased and was considerably higher in the 9-year-old alfalfa field than in other treatments (*p* < 0.05) (Figure 4).

### 3.5. Soil Nematode Metabolic Footprint and Carbon Flow Within the Food Web

The metabolic footprints of fungivorous nematodes in the CK were substantially higher than those in the alfalfa fields (Figure 5a, *p* < 0.05). The metabolic footprints of plant-parasitic nematodes were significantly higher in the 2-year alfalfa fields than in the 9-year and 18-year alfalfa fields (*p* < 0.05). The metabolic footprint of omnivorous-predation nematodes in 9-year and 18-year alfalfa fields notably surpassed that of other treatments (*p* < 0.05). However, neither the total nor the bacterivorous nematode metabolic footprint showed any apparent changes. In comparison to 2-year and 18-year alfalfa fields, the functional metabolic footprint (total area) of nematodes in 9-year alfalfa fields was higher (Figure 5b). The CK and 2-year treatments were in the A and B quadrants, the 9-year treatment was in the B quadrant, and the 18-year treatment was in the B and C quadrants. The results indicate that 9-year alfalfa fields exhibit the strongest capacity to regulate the soil food web, with the most stable food web structure. Additionally, the 9-year treatment demonstrated the highest proportion of energy flow through bacterial and fungal channels and the lowest proportion through plant energy channels. This suggests that energy conversion and utilization efficiency within the soil food web of the 9-year alfalfa fields was relatively high (Figure 5d).

As shown in Figure 5c, Plant root nutrient levels directly influenced the fungal community structure, plant-parasitic nematodes’ carbon footprint, and omnivorous-predatory nematodes’ carbon footprint (*p* < 0.05). Additionally, the carbon footprint of omnivorous-predatory nematodes was found to have a direct impact on soil organic carbon accumulation (*p* < 0.05). Alfalfa planting duration had no significant effect on bacterial communities or the carbon footprint of bacterivorous nematodes. However, the fungal community structure directly influenced the carbon footprint of fungivorous nematodes (*p* < 0.05). The model accounted for 79.0% of the variation in the carbon footprint of omnivorous-predatory nematodes and 3.0%, 37.0%, and 68.0% of the variation in carbon metabolism of bacterivorous, fungivorous, and plant-parasitic nematodes, respectively. Additionally, the model explained 19.0%, 66.0%, and 53.0% of the variation in bacterial community structure, fungal community structure, and soil organic carbon, respectively.

### 3.6. Symbiotic Relationship Between Soil Microorganisms and Nematodes

As shown in Figure 6, Among the alfalfa fields with varying planting durations, the 2-year and 18-year fields exhibited higher total and positive link counts than other treatments. In contrast, the 9-year-old fields displayed fewer positive links and more negative links than other treatments. Each node’s size is inversely proportional to its centrality score, and the node with a more prominent centrality is the critical species of the network. For the CK treatment, key species include *Leptosphaeria*, *Articulospora*, *Acremonium*, *Alternaria*, and *Pyrenochaeta*. For the 2-year treatment, they are *Massilia* and *Volvariella*; for the 9-year treatment, they include *Acrobloides*, *Mesorhabditis*, *Pir4_lineage*, *Articulospora*, *Alaimus*, *Basiria*, *Lecythophora*, *Isolaimium*, *Meloidogyne*, and *Pterula*; and for the 18-year treatment, they are *Solicoccozyma*, *Acrobloides*, *Aphenlenchus*, *Aulolaimus*, *Eucephalobus*, *Basiria*, *Meloidogyne*, *Clitopilus*, *Leohumicola*, *Gibberella*, *Chiloplacus*, and *Anguina*. Consequently, key species evolve as planting years change. Furthermore, network analysis revealed that in the CK and 18-year treatments, fungi, fungivorous nematodes (FF), and omnivorous-predatory nematodes (OP) were positively correlated, indicating that fungi primarily drive the degradation pathways in the soil food web. In contrast, in the 2-year and 9-year treatments, a positive relationship was observed between bacteria, bacterivorous nematodes (BF), and OP, suggesting that bacteria dominate the degradation processes in these food webs.

### 3.7. Analysis of the Correlation Between Soil Microbial and Nematode Communities and Environmental Factors

We used genus-level data of soil microorganisms and nematode communities as response variables and soil physicochemical properties as explanatory variables in a redundancy analysis (Figure 7). The redundancy analysis of environmental variables and nematode communities revealed that *Hoplolaimus*, *Pratylenchus*, *Aglenchus*, *Filenchus*, *Basiria*, and *Aphenlenchus* were positively correlated with total phosphorus (TP), available phosphorus (AP), and soil moisture (SM). In contrast, *Isolaimium*, *Paratylenchus*, *Acrobeles*, *Helicotylenchus*, and *Aporcelaimellus* were positively correlated with soil organic carbon (SOC), total nitrogen (TN), bulk density (BD), and pH. TN and AP emerged as the dominant factors influencing the composition of nematode communities in this study. Similarly, analysis of soil bacterial communities showed that *RB41*, *Massilia*, *Arthrobacter*, *Microvirga*, *Nocardioides*, and *Blastococcus* were positively correlated with pH, TN, BD, and SOC. Meanwhile, *Subgroup_10*, *MND1*, *Nitrospira*, *Pir4_lineage*, *Pirellula*, and *Bacillus* were positively correlated with SM, TP, and AP. As with the nematode communities, TN and AP were the predominant factors affecting bacterial community composition. For the soil fungal communities, *Metarhizium*, *Entoloma*, *Setophoma*, *Chaetomium*, and *Glarea* were positively correlated with SM, AP, and TP, while *Paraphoma*, *Lectera*, and *Cyphellophora* were positively correlated with BD, pH, SOC, and TN. In this case, TN and AP played a critical role in shaping the composition of fungal communities.

## 4. Discussion

### 4.1. Effects of Different Planting Years on Soil Microorganisms and Nematodes

This study focuses on the interactions between communities of nematodes and soil microorganisms in alfalfa fields with various planting years. Generally speaking, the extension of alfalfa planting years will accelerate the degradation of vegetation. However, an increasing number of studies indicate that plants only measure degradation and that soil microorganisms and nematode communities offer insights into the functioning of alfalfa ecosystems over different planting years [45]. Different planting years significantly changed soil microorganisms’ and nematode communities’ diversity and richness [46]. In this study, the diversity of soil nematodes increased significantly with the extension of planting years, indicating that the 9-year and 18-year alfalfa fields made the soil nematode community more diverse and structured (Figure 5b). Additionally, compared to other planting years, the 9-year alfalfa field had a much higher diversity of soil microorganisms (Figure 4). Extending the planting years has been shown to increase soil microbial diversity and subsurface biomass [47], which improves soil resources and benefits the soil microenvironment and soil nematode communities [48]. Therefore, in the semi-arid areas of the Loess Plateau, soil health and nutrient cycles can be optimized by rationally planning the planting years. PCA showed that the composition of soil microorganisms and nematode communities had significant differences with increasing planting years, findings supported by other studies [14,49]. When alfalfa is grown for an extended period, soil nutrients and organic carbon continuously accumulate. According to earlier studies, changes in soil microbial and nematode communities are significantly influenced by the physicochemical characteristics of the soil and the biological environment [50]. Redundancy analysis (RDA) indicated that soil TN and AP primarily influence the dynamics of soil nematodes and microbial communities (Figure 7). Nitrogen and phosphorus play pivotal roles in regulating soil fauna and notably impact microbial communities [51]. Therefore, in agricultural practices on the semi-arid Loess Plateau, nitrogen and phosphorus fertilizers should be applied judiciously to support the growth of beneficial microorganisms and nematodes while minimizing resource waste and preventing environmental pollution from over-fertilization. Nitrogen meets the needs of plant growth, and copious root exudates have an impact on the microbial populations in the soil [52]. Similar to the findings of Cleveland et al. [53], phosphorus becomes the primary environmental factor restricting the organization of soil microbial communities since it is difficult to move in the soil and be consumed by crops. Alfalfa abundance is closely related to soil microorganisms and nematode communities, and root exudates are a source of carbon and energy available to microorganisms. The rhizosphere is more conducive to microbial reproduction because unstable chemicals produced by roots can boost microbial growth and activity. Bacterivorous and fungivorous nematodes enhance plant nutrient uptake by preying on microorganisms [54]. Omnivorous-predatory nematodes, by consuming microvorous nematodes at lower trophic levels, indirectly affect bacterial and fungal community compositions, affecting plant growth. Moreover, by preying on plant-parasitic nematodes, they reduce plant diseases and insect infestations [55]. These findings support that the interaction among soil microorganisms, soil properties, and vegetation abundance offers a comprehensive metric for assessing soil health [56]. Protecting and promoting the growth of beneficial soil microorganisms and nematodes can reduce the incidence of plant diseases and pests, enhancing the sustainability of agricultural production. The interaction network between nematodes and soil microorganisms can prove the principle of microbial and nematode community assembly, offer insights into the interactions of various groups under natural conditions, and identify key groups with the greatest influence on community structure [57]. Degradation of grasslands generally intensifies the interactions between nematode communities and soil microbes, leading to a more intricate network structure. Although a network’s stability was directly correlated with its complexity, overly complex systems might jeopardize an ecosystem’s stability. Consequently, microbial and nematode communities exposed to environmental stress often exhibit reduced network stability [58]. In this study, the 2-year and 18-year alfalfa fields exhibited a stronger positive correlation between soil microbial and nematode community species (Figure 6). This may be due to reduced antagonism between soil microbial and nematode communities and increased environmental filtering under evolving environmental stress, which led to a decrease in the proportion of both negatively and positively correlated species aggregations under these conditions. The interactions between soil microbial and nematode community species in the 9-year alfalfa fields exhibited an opposite trend. This aligns with findings from other studies, which suggest that negative correlations enhance the stability of ecological networks under disturbance. Negative correlations may act as balancing mechanisms, promoting resilience by preventing any one group from becoming overly dominant, thereby contributing to the overall stability of the network in the face of environmental stressors [55,59]. When the external environment is disturbed, the environmental disturbance can rapidly transmit to the entire network, resulting in an unstable network structure [60]. Such instability in the microbial and nematode network can result in significant changes in the communities responsible for soil carbon and nitrogen nutrient cycling, thereby influencing the overall function of grassland ecosystems [61].

### 4.2. Effects of Different Planting Years on Soil Food Web Metabolic Activities and Carbon Flow

The metabolic footprint of nematodes represents the nematode population’s metabolic activity and their habitat ecosystem’s functional state. Additionally, it offers valuable insights into carbon metabolism pathways and energy flow within the food web, making it an effective indicator for evaluating the functionality of regional food webs. Enrichment footprints can quickly respond to the metabolic activity of lower nematode populations in resource accumulation, and structural footprints often reflect the net resource output of the food web and the metabolic activity of nematodes at high trophic levels [12,13]. In this study, the enrichment footprint in the 9-year and 18-year alfalfa fields was notably lower compared to the 2-year fields (Figure 5b). Conversely, the structural footprint in the 9-year and 18-year fields surpassed that of the 2-year fields. These observations suggest that the metabolic activity of omnivorous-predatory nematodes in the 9-year and 18-year fields was significantly more potent than in the 2-year fields. Additionally, the metabolic footprint of plant-parasitic nematodes was diminished considerably in the 9-year and 18-year fields relative to the 2-year fields. This indicated that as planting years extend, the metabolic activity of harmful nematodes is inhibited, enhancing the activity of beneficial nematodes and thereby improving the stability of the soil ecosystem. The reason for these differences was the annual accumulation of nitrogenous compounds produced by alfalfa, which inhibits plant-parasitic nematodes. At the same time, the extension of alfalfa planting years results in augmented organic matter input to the soil, strengthening the productivity and metabolic activity of lower-trophic-level prey, thereby creating favorable conditions for the reproduction and metabolism of predators and maintaining the metabolic balance of the system [14,62]. This approach can foster healthier soil ecosystems and improve the resistance and resilience of agricultural systems. The enrichment and structural footprints define the functional metabolic footprint. This metric can be used to measure the ability of the food web to self-regulate and maintain metabolic balance. A more considerable value indicates the nematode population’s more vital regulatory ability over the habitat’s food web. Additionally, it suggests the enhanced capability of predators and prey to sustain their metabolic balance, leading to increased carbon inputs into the soil micro-food web [11,14]. The study demonstrates that during the growth of alfalfa, the functional metabolic footprint of nematodes in the 9-year alfalfa field surpasses that of the 2-year and 18-year fields. Consequently, there is a higher carbon input into the soil micro-food web during the 9-year field. As alfalfa grows, external resources are consistently added to the soil food web, offering an optimal habitat for soil microorganisms. Physicochemical factors such as SOC and TN in the 18-year alfalfa field were significantly higher than those in other treatments. This led to the evolution of higher-quality organic phosphorus, a more balanced food web metabolism, and enhanced resource enrichment in the nematode-inhabited soil ecosystem [63]. As the ecosystem’s population evolves, omnivorous-predatory nematodes diminish the count of nematodes at lower trophic levels through established trophic relationships [13]. The findings indicate a significant increase in the metabolic footprint of omnivorous-predatory nematodes after 9 years of alfalfa cultivation, while the metabolic footprints of nematodes from other taxa decreased. In addition, soil nematode faunal analysis and food web energy flow analysis revealed that the soil food web was more stable after 9 years of alfalfa growth. However, the alfalfa field was degraded when alfalfa was grown to 18 years old. Prior research has also suggested that the soil food web during the mid-forest stage remains more stable than in the young or old forest stages [14]. This phenomenon can be explained by the “intermediate disturbance hypothesis”, in which the intensity of external disturbance is maintained at a medium level and the ecosystem is more stable [64]. Therefore, prolonged alfalfa cultivation can lead to increased surface nutrient absorption, thereby contributing to soil ecosystem degradation.

Decomposition pathways within the soil food web significantly influence the rate of soil carbon loss. Notably, while the bacterial decomposition pathway facilitates rapid nutrient turnover, it is less effective than the fungal decomposition pathway in sequestering soil carbon and nitrogen [60]. The alfalfa root system is rich in nutrients, and the quality of these nutrients affects the transformation of bacterial and fungal decomposition pathways in the soil food web, thereby affecting carbon turnover efficiency [65]. Specifically, bacterial decomposition pathways predominate in decomposing quickly decomposable carbon resources, whereas fungal pathways dominate in breaking down difficult-to-decompose carbon sources [66]. In our study, fungal decomposition pathways were stronger than bacterial ones in increasing soil organic carbon. Prior research indicates that fungi play a more prominent role than bacteria in protecting carbon pools, enhancing the physical environment for carbon stability, and preventing the accumulation of microbial-derived organic carbon [67]. The decomposition pathways of bacteria and fungi within the soil food web are not separated. They operate concurrently, featuring varying carbon transfer and utilization processes [68]. In alfalfa fields with different planting years, the amounts of carbon resource inputs distinctly influence the composition of microbial and nematode communities and their respective decomposition pathways. In addition, the impact of environmental and climatic conditions on the food web cannot be ignored. Therefore, further research is essential to comprehend the role of soil bacteria and fungi in the food web during alfalfa fields with different planting years.

### 4.3. Ecological Impact and Limitations of This Study

This study examined the soil micro-food web, utilizing nematode faunal analysis and soil food web connectivity and energy flow networks to explore the effects of planting duration on the structure, function, and decomposition pathways of soil micro-food webs in alfalfa fields. The 9-year alfalfa planting treatment exhibited the highest energy conversion and utilization efficiency within the soil food web, with the most stable structure. Therefore, planting alfalfa for 9 years can effectively improve soil health. This duration increases soil organic matter content, enhancing soil water retention capacity and nutrient cycling efficiency [69]. These improvements promote the activity and diversity of soil microorganisms and nematodes, forming a more stable and efficient soil food web [70]. Additionally, 9-year alfalfa planting aids in carbon fixation and reduces greenhouse gas emissions, which is significant for addressing global climate change [71].

In recent years, no studies have evaluated the carbon storage of soil micro-food webs in alfalfa fields. During the global plant growing season, soil micro-food web organisms’ average monthly carbon turnover reached 0.14 Gt, with 0.11 Gt released into the atmosphere through respiration. The carbon released by the respiration of soil micro-food web organisms is equivalent to 15% of the carbon released by fossil fuel combustion and 2.2% of the total soil CO_2_ release. Therefore, soil micro-food webs are crucial in the global carbon cycle [50,72]. Studies have shown that alfalfa planting for more than two years significantly increases soil organic carbon storage. Planting alfalfa for 9 and 18 years enhances soil organic matter as the roots die and renew, potentially growing crop yields by promoting soil biological functions and nutrient transport through residue decomposition [73].

The results of this study offer farmers and policymakers valuable insights into the effective management of alfalfa cultivation in the context of climate change, highlighting the crucial role of micro-food webs in enhancing soil health and crop productivity. Effective management of micro-food webs can improve the sustainability of agricultural systems. However, achieving these goals requires long-term research, rational resource allocation, and comprehensive policy support. Additionally, the sampling plot design in this study may introduce some bias. Since the plots were selected in specific geographic locations, they may be constrained by regional environmental conditions, such as climate and soil type, potentially influencing the results. Future studies should replicate the experiments across multiple locations to confirm the generalizability of the findings. Furthermore, this study’s division of planting years may need to be simplified. Although this study included plots with planting durations of 2, 9, and 18 years, actual applications may require a broader range of planting years. Future research could consider a finer gradient of planting years to more accurately capture changes in soil microorganisms and nematode communities in alfalfa fields. This study mainly relied on laboratory analysis, which may not fully represent the soil ecosystem. Future research could integrate field observations and long-term monitoring with laboratory analysis to more accurately describe the effects of planting duration on soil microorganisms and nematode communities in alfalfa fields. A comprehensive approach also allows for considering other factors that impact soil ecosystems, exploring the adaptability of soil microorganisms and nematode communities to environmental changes.

## 5. Conclusions

Prolonged alfalfa cultivation in the semi-arid Loess Plateau significantly increased the richness and diversity of soil microorganisms and nematode communities, leading to significant structure changes. Plant root periphery nutrients directly influence the fungal community structure and the carbon metabolism of plant-parasitic and omnivorous-predatory nematodes. Furthermore, the carbon metabolism of omnivorous-predatory nematodes directly affects soil organic carbon sequestration. As alfalfa planting years extend, the metabolic footprint of plant-parasitic nematodes decreases while that of omnivorous-predatory nematodes increases. In addition, among different alfalfa planting years, the 9-year treatment exhibits the highest energy conversion and utilization efficiency within the soil food web, maintaining the most stable structure. In the decomposition pathway of the soil micro-food web, the alfalfa planting years promote the conversion between bacteria and fungi and then stimulate the conversion between BF and FF. The 9-year-old alfalfa field mainly relies on bacterial decomposition pathways. The trophic interactions between soil microbial and nematode communities primarily drive carbon flow through decomposition pathways.

## Figures and Tables

**Figure 1 microorganisms-12-02268-f001:**
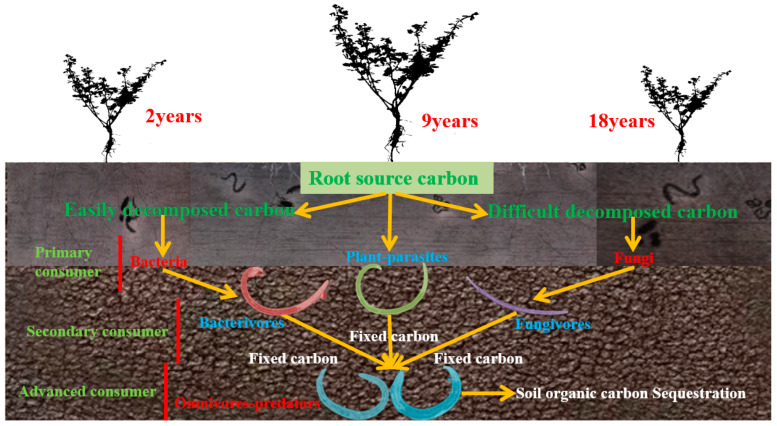
A conceptual framework illustrates how alfalfa inputs carbon into the soil through its roots at various growth stages. Microorganisms decompose and utilize this carbon, with a portion ultimately fixed in the soil as organic carbon sequestration. This process involves microbial activities and interactions among consumers at different trophic levels. Together, these actions sustain the health and sustainability of the soil ecosystem.

**Figure 2 microorganisms-12-02268-f002:**
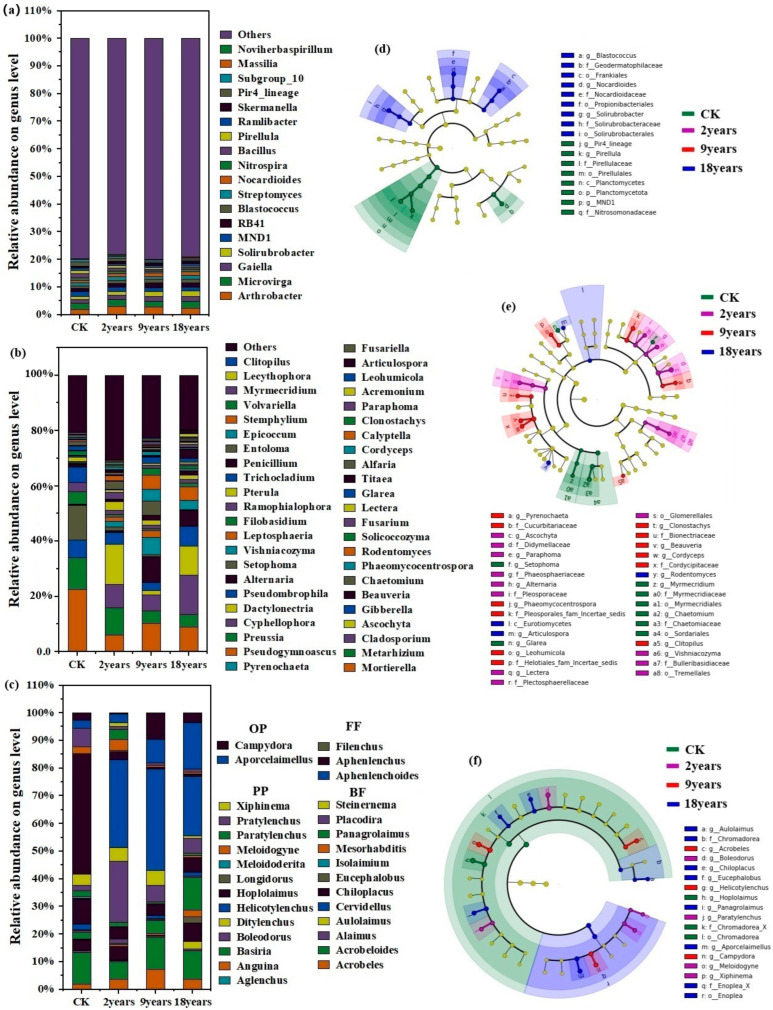
Composition and difference analysis of soil microorganisms and nematode communities in alfalfa fields with different alfalfa planting years (on genus level), (**a**): relative abundance of bacterial communities; (**b**): relative abundance of fungal communities; (**c**): relative abundance of nematode communities; (**d**): Lefse analysis of the soil bacteria community; (**e**): Lefse analysis of the soil fungal community; (**f**): Lefse analysis of the soil nematode community. PP—plant parasites; BF—bacterial predators; FF—fungal predators; OP—omnivorous predators.

**Figure 3 microorganisms-12-02268-f003:**
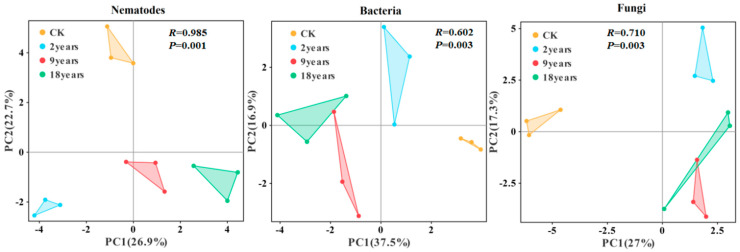
Principal component analysis (PCA) of bacterial, fungal, and nematode communities based on Bray–Curtis distances.

**Figure 4 microorganisms-12-02268-f004:**
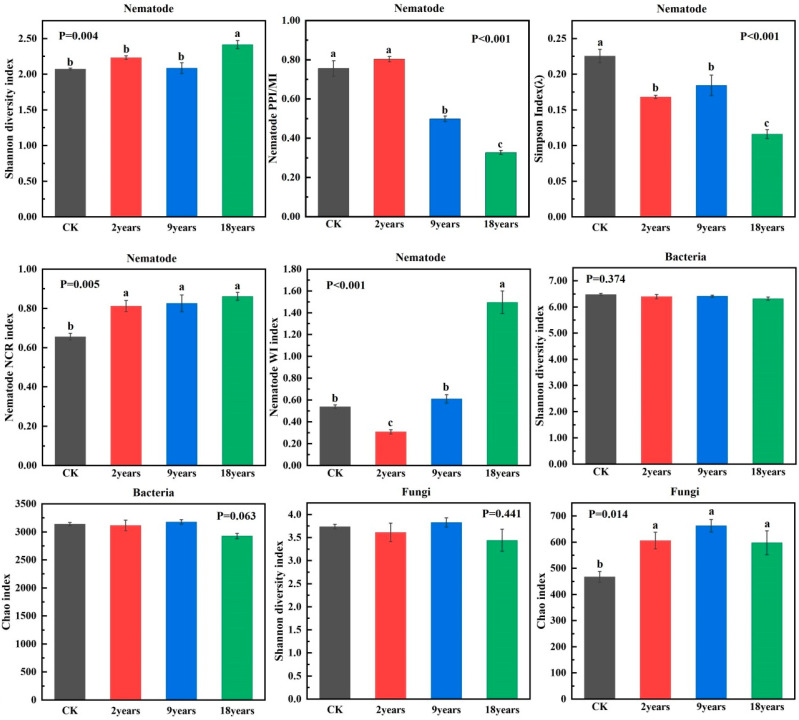
Ecological function index of soil microorganisms and nematode communities. Different lowercase letters indicate significant differences with a *p* value < 0.05 based on the analysis of variance.

**Figure 5 microorganisms-12-02268-f005:**
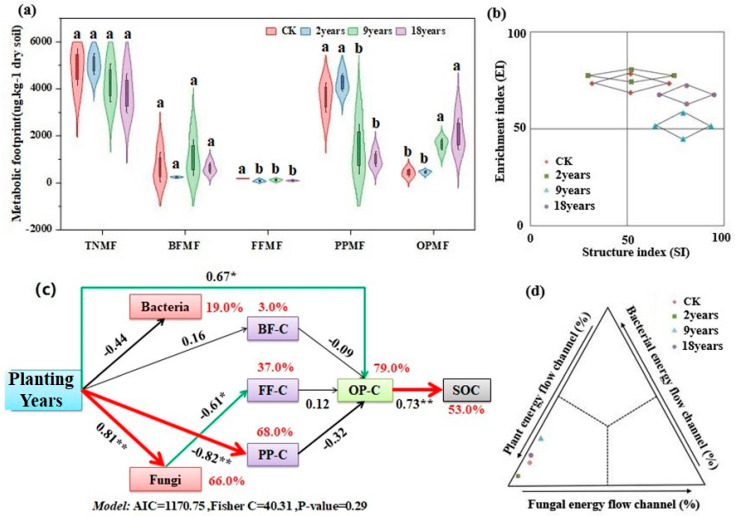
Soil nematode metabolic footprint and food web carbon flow. (**a**) Soil nematode metabolic footprint, (**b**): soil nematode fauna analysis, (**c**): micro-food web decomposition pathway analysis, (**d**): food web carbon flow analysis. Functional metabolic footprint consists of sequential connection points: (SI-0.5Fs/k, EI), (SI + 0.5Fs/ k, EI), (SI, EI-0.5Fe/k) and (SI, EI + 0.5Fe/k), the k value is 100. BF-C: bacterivorous nematode carbon footprint; FF-C: fungivorous nematode carbon footprint; PP-C: plant-parasitic nematode carbon footprint; OP-C: omnivorous-predation nematode carbon footprint; SOC: soil organic carbon. The red solid arrow indicates the significant difference (*p* < 0.01), the green solid arrow indicates the significant difference (*p* < 0.05), and the black solid arrow indicates the difference is not significant (*p* > 0.05). The asterisk indicated a significant difference between treatments (* *p* < 0.05, ** 0.01 *< p* < 0.05). Different lowercase letters indicate significant differences with a value of *p* < 0.05 based on the ANOVA.

**Figure 6 microorganisms-12-02268-f006:**
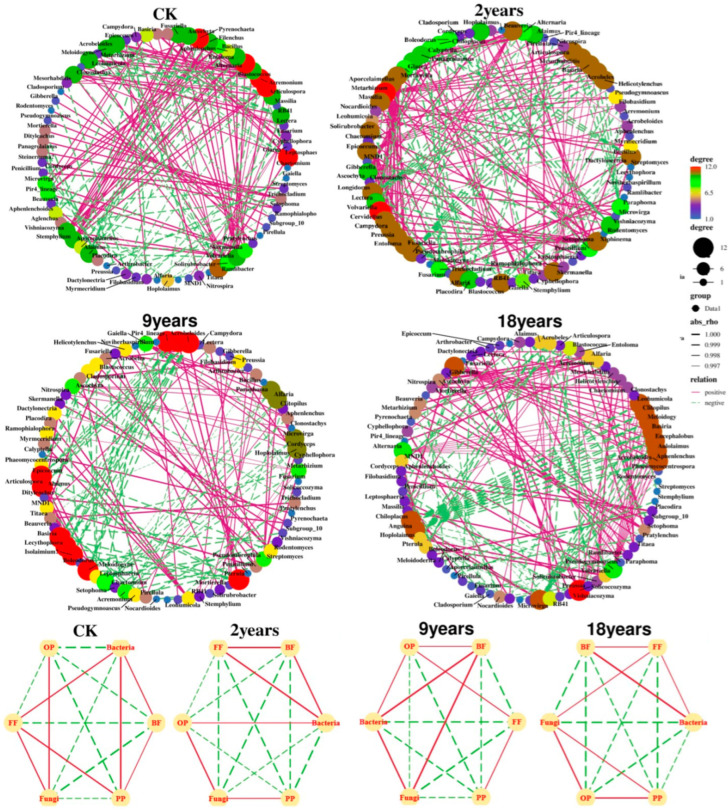
Network visualization of the interaction intensity of soil micro-food webs in alfalfa fields with different planting years, the size of each node is proportional to the centrality score, and genera with larger centrality represent the key species of each network. The line between each pair of nodes represents a positive (red) or negative (dashed green) interaction, and the line’s thickness indicates the correlation’s strength. BF: bacterivorous nematodes; FF: fungivorous nematodes; PP: plant-parasitic nematodes; OP: omnivorous-predatory nematodes.

**Figure 7 microorganisms-12-02268-f007:**
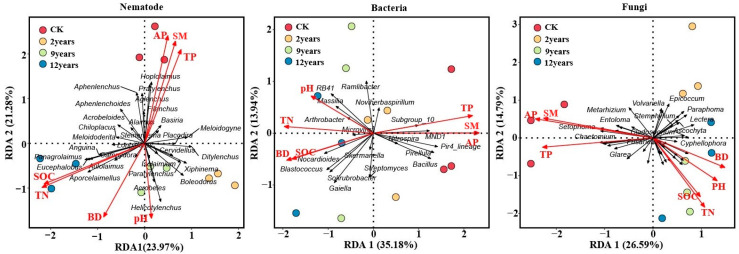
Correlation analysis between soil microorganism and nematode communities and soil physicochemical factors. SM: soil moisture content; SOC: soil organic carbon; TN: total nitrogen; TP: total phosphorus; AP: available phosphorus; BD: bulk density.

## Data Availability

The microbial and nematode DNA sequences from the 12 soil samples have been deposited in the Sequence Read Archive (SRA) of the NCBI database under Accession numbers NCBI: PRJNA1010171 and SRA: SUB13801684.

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
