# Peer review of "Effects of Perennial Alfalfa on the Structure and Function of Soil Micro-Food Webs in the Loess Plateau"

_microorganisms, 2024, doi:10.3390/microorganisms12112268_

Round 1
Reviewer 1 Report
Comments and Suggestions for Authors
The paper provides valuable insights into how different alfalfa planting years affect the soil micro-food web and carbon sequestration, focusing on the Loess Plateau, a critical ecological area in China. The research methodology is thorough, using high-throughput sequencing and nematode morphological analysis to understand soil microbial and nematode community dynamics. The combination of various analytical tools (PCA, SEM, and network analysis) adds depth to the findings, and the results are significant for both soil health and agricultural management. The paper is of merit, but some key areas need improvement to enhance clarity and impact.
1. Abstract provides a solid overview, but it could be more concise. Reducing redundancy, particularly in stating results, would improve readability.
2. The introduction does well in framing the study, but additional context about the practical implications of alfalfa planting on local agriculture and its broader significance in carbon management would strengthen the argument. The role of the Loess Plateau in global carbon sequestration could be further elaborated.
3. The experimental design section is clear but could benefit from more detail on why specific treatments (2, 9, and 18-year alfalfa fields) were selected. Justification for this choice would strengthen the rationale.
4. The section on DNA extraction and high-throughput sequencing lacks specific reference to quality controls. Adding details on error reduction and sequencing bias minimization would improve scientific rigor.
5. The statistical methods (e.g., ANOVA, SEM) are appropriate, but the reasoning for using these specific analyses in this context could be better explained, especially for readers unfamiliar with these techniques.
6. The results are robust, but there are too many complex findings presented in rapid succession, making it difficult for readers to follow. Breaking down the results into more digestible sections, with clear subheadings, would enhance clarity. Simplify the presentation of the PCA, SEM, and network analyses for a broader audience, including ecologists and agricultural scientists who may not be experts in these methods.
7. The paper lacks a discussion on limitations and potential biases. Addressing limitations such as the specific environmental conditions of the study site and the lack of replication across other regions would lend credibility.
8. How could these findings inform crop rotation strategies, and what role could climate change play in altering the effects of alfalfa planting?
9. Provide a more in-depth discussion on how these findings can be applied to improve agricultural sustainability, particularly in other semi-arid regions facing similar challenges.
Author Response
The paper provides valuable insights into how different alfalfa planting years affect the soil micro-food web and carbon sequestration, focusing on the Loess Plateau, a critical ecological area in China. The research methodology is thorough, using high-throughput sequencing and nematode morphological analysis to understand soil microbial and nematode community dynamics. The combination of various analytical tools (PCA, SEM, and network analysis) adds depth to the findings, and the results are significant for both soil health and agricultural management. The paper is of merit, but some key areas need improvement to enhance clarity and impact.
- Abstract provides a solid overview, but it could be more concise. Reducing redundancy, particularly in stating results, would improve readability.
Reply: Thank you for your valuable comments. Regarding the conciseness of the Abstract and avoiding duplication, we will try to reduce the redundancy of the Results section in the revised version. Please check lines 26-40.
- The introduction does well in framing the study, but additional context about the practical implications of alfalfa planting on local agriculture and its broader significance in carbon management would strengthen the argument. The role of the Loess Plateau in global carbon sequestration could be further elaborated.
Reply: Thank you for your comments. We have revised the Introduction. Please check lines 89-98 and Lines101-104.
- The experimental design section is clear but could benefit from more detail on why specific treatments (2, 9, and 18-year alfalfa fields) were selected. Justification for this choice would strengthen the rationale.
Reply: Thank you for your valuable comments. We have detailed the reasons for choosing these treatments in the revised manuscript. Please check lines170-174.
- The section on DNA extraction and high-throughput sequencing lacks specific reference to quality controls. Adding details on error reduction and sequencing bias minimization would improve scientific rigor.
Reply: Thank you for your valuable comments. We have described the steps and process of high-throughput sequencing in detail in the revised manuscript. Please check lines 203-217.
- The statistical methods (e.g., ANOVA, SEM) are appropriate, but the reasoning for using these specific analyses in this context could be better explained, especially for readers unfamiliar with these techniques.
Reply: Thank you very much for your suggestion, we have revised the statistical methods section in detail according to your suggestion. Please check lines 275-289.
- The results are robust, but there are too many complex findings presented in rapid succession, making it difficult for readers to follow. Breaking down the results into more digestible sections, with clear subheadings, would enhance clarity. Simplify the presentation of the PCA, SEM, and network analyses for a broader audience, including ecologists and agricultural scientists who may not be experts in these methods.
Reply: Thank you for your valuable comments. We have described and analyzed the Results section in detail in the revised manuscript. Please check the blue word marks in the Results.
- The paper lacks a discussion on limitations and potential biases. Addressing limitations such as the specific environmental conditions of the study site and the lack of replication across other regions would lend credibility.
Reply: Thank you very much for your suggestion. We have added this part in the revised manuscript. See lines 576-590 for details.
- How could these findings inform crop rotation strategies, and what role could climate change play in altering the effects of alfalfa planting?
Reply: Thank you for your question. Long-term planting of the same crop can lead to reduced diversity in soil microbial communities, while crop rotation increases microbial diversity and enhances soil ecosystem stability and health. Monitoring changes in microbial communities helps farmers assess soil health and adjust rotation plans as needed. Additionally, different crops have varying effects on soil nutrient consumption and replenishment. Crop rotation balances nutrient levels in the soil, preventing the depletion or excessive accumulation of specific nutrients. Since microorganisms are integral to nutrient cycling, shifts in microbial communities can offer valuable insights for farmers in managing soil nutrients more effectively.
Climate change can introduce temperature fluctuations that impact alfalfa’s growth cycle and yield. Alfalfa has a limited temperature tolerance, and exceeding this range may adversely affect its growth. Alfalfa also requires optimal water levels; excessive moisture can cause root diseases, while insufficient rainfall reduces growth and yield. Climate change may alter precipitation patterns, directly influencing alfalfa growth. Additionally, climate change can modify pest and disease cycles and distributions, potentially leading to new or intensified pest and disease pressures.
Therefore, in developing crop rotation strategies, it is essential to consider climate change impacts on alfalfa cultivation and adopt adaptive measures. This may include selecting alfalfa varieties more resilient to climate change or incorporating crops with higher tolerance to extreme weather events into the rotation plan. Employing the latest scientific research and adaptive agricultural technologies can support sustainable and efficient agricultural production.
- Provide a more in-depth discussion on how these findings can be applied to improve agricultural sustainability, particularly in other semi-arid regions facing similar challenges.
Reply: Thank you for your valuable comments. This paper studied the interaction between nematode communities and soil microorganisms in alfalfa fields under different planting years, explored the interactions between soil microorganisms, nematodes, soil properties and vegetation abundance, and the effects of these interactions on soil health and ecosystem functions. In order to optimize soil health and nutrient cycles and improve the resistance and resilience of agricultural systems, in agricultural practices in the semi-arid areas of the Loess Plateau, the planting years should be reasonably planned, nitrogen and phosphorus fertilizers should be reasonably applied, and the growth of beneficial soil microorganisms and nematodes should be protected and promoted. At the same time, the role of soil bacteria and fungi in the food web of alfalfa fields with different planting years was studied in depth, in order to provide a scientific basis for agricultural production. Please check lines 437-441, lines 447-453, lines 461-466, lines 501-514.

Reviewer 2 Report
Comments and Suggestions for Authors
This manuscript presents very interesting research results on environmental factors and on the interactions between communities of nematodes and soil microorganisms in alfalfa fields with various planting years. The experiment was conducted on alfalfa grassland, which was converted from farmland and sown in 2003 (18 years), 2012 (9 years), and 2019 (2 years) in northwest China. Control conditions were provided. The introduction very well illustrates the problem to which this manuscript is devoted. The aim of this work is clearly specified: i) assess the impact of different planting durations on soil microbial and nematode community structures, ii) analyze the influence of planting years on decomposition pathways within the soil micro-food web and iii) identify optimal planting durations for enhancing soil fertility and ecosystem stability. The methodology is very well and thoroughly presented. According to standard methods, the following were studied: pH of the soil, soil organic carbon and nitogen, total phosphorus, soil-available phosphorus, nitrate nitrogen, and ammonium nitrogen. Fungi and bacteria were studied by DNA extraction and High-Throughput Sequencing. Their analysis was done for genera levels. Nematodes after extraction from soil were identified using a microscope. The results were presented in great detail and clearly using well-thought-out Figures. The results are very interesting and bring many new elements to science and acricultural praxis. The manuscript is prepared very carefully. However, the Authors should analyze whether they use the terms ‘flora’ and ‘nematode flora’ correctly in the text. This is unclear. Besides, the title is too narrow and overemphasizes only the problem of carbon sequestration. The goals of the current work are much broader. After correction of a few minor errors (see Remarks) the manuscript should be published in Microorganisms/MDPI.
Remarks
Line 206 Ferris et al (2006) – the number from References should be given
Line 246 2.7. Nematode Metabolic Footprint and Flora Analysis – it is not clear what the authors mean by Flora analysis
Line 312 not all abbreviations used in Figure 2 are explained
Line 319 (Fig.3) a space should be added before 3
Line 417 Figure 7 should be cited here
Line 448 (Fig5b) – this needs to be changed
Line 450 microorganisms(Fig 4). – this needs to be changed. Such changes should also be made in other places in Discussion
Line 571 nematode flora – this expression is incorrect and should be corrected
Line 592 decomposition[68]. – this needs to be changed
Line 724 Pinus tabulaeformis – it should be in italic
Literature – it is partly not written in accordance with the recommendations of the MDPI editorial office
Author Response
Comments and Suggestions for Authors
This manuscript presents very interesting research results on environmental factors and on the interactions between communities of nematodes and soil microorganisms in alfalfa fields with various planting years. The experiment was conducted on alfalfa grassland, which was converted from farmland and sown in 2003 (18 years), 2012 (9 years), and 2019 (2 years) in northwest China. Control conditions were provided. The introduction very well illustrates the problem to which this manuscript is devoted. The aim of this work is clearly specified: i) assess the impact of different planting durations on soil microbial and nematode community structures, ii) analyze the influence of planting years on decomposition pathways within the soil micro-food web and iii) identify optimal planting durations for enhancing soil fertility and ecosystem stability. The methodology is very well and thoroughly presented. According to standard methods, the following were studied: pH of the soil, soil organic carbon and nitogen, total phosphorus, soil-available phosphorus, nitrate nitrogen, and ammonium nitrogen. Fungi and bacteria were studied by DNA extraction and High-Throughput Sequencing. Their analysis was done for genera levels. Nematodes after extraction from soil were identified using a microscope. The results were presented in great detail and clearly using well-thought-out Figures. The results are very interesting and bring many new elements to science and acricultural praxis. The manuscript is prepared very carefully. However, the Authors should analyze whether they use the terms ‘flora’ and ‘nematode flora’ correctly in the text. This is unclear. Besides, the title is too narrow and overemphasizes only the problem of carbon sequestration. The goals of the current work are much broader. After correction of a few minor errors (see Remarks) the manuscript should be published in Microorganisms/MDPI.
Reply: Thank you for your valuable comments. We have revised the title to " Effects of Perennial Alfalfa on the Structure and Function of Soil Micro-Food Webs in the Loess Plateau ".
2.Line 206 Ferris et al (2006) – the number from References should be given
Reply: Thank you for your valuable comments. Regarding your question about the reference number, I have checked and confirmed that all references to Ferris et al. (2006) in the article have been correctly marked with the reference number.
3.Line 246 2.7. Nematode Metabolic Footprint and Flora Analysis – it is not clear what the authors mean by Flora analysis
Reply: Thank you very much for your comments. We have carefully reviewed the information and have changed "Flora Analysis" to " faunal analysis ". The faunal analysis of nematodes is mainly to reflect the resistance of the soil food web, the response to external disturbances and the evaluation of the changes in food resources in the soil environment. Ferris et al. (2001) combined the structural index (SI) and the enrichment index (EI) as the coordinate site center site, and used the enrichment metabolic footprint (Fe) and the structural metabolic footprint (Fs) to draw diamonds in the quadrant. The area of ​​the diamond is the functional metabolic footprint of nematodes. The faunal analysis can well explain the differences between carbon metabolism indicators in different planting years.
4.Line 312 not all abbreviations used in Figure 2 are explained
Reply: Thank you for your valuable comments. We have explained all the abbreviations in the revised manuscript. Please check lines 316-317.
5.Line 319 (Fig.3) a space should be added before 3
Reply: Thank you for your valuable feedback. We have revised it. Please check lines 320 and 329.
6.Line 417 Figure 7 should be cited here
Reply: Thank you for your valuable feedback. We have revised it. Please check line 407.
7.Line 448 (Fig5b) – this needs to be changed
Reply: Thank you for your valuable feedback. We have revised it. Please check lines 352, 436 and 497.
8.Line 450 microorganisms (Fig 4). – this needs to be changed. Such changes should also be made in other places in Discussion
Reply: Thank you for your valuable feedback. We have revised it. Please check lines437.
9.Line 571 nematode flora – this expression is incorrect and should be corrected
Reply: Thank you very much for your careful review and valuable comments. We have revised it. Please check lines 553.
10.Line 592 decomposition [68]. – this needs to be changed
Reply: Thank you very much for your feedback, we have rewritten the sentence. Please check lines 544-546.
11.Line 724 Pinus tabulaeformis – it should be in italic
Reply: Thank you very much for your careful review and valuable comments. We have revised it. Please check lines733.
12.Literature – it is partly not written in accordance with the recommendations of the MDPI editorial office
Reply: Thank you very much for your feedback. We have re-edited the references to follow the formatting requirements of the MDPI editorial office.

Round 2
Reviewer 1 Report
Comments and Suggestions for Authors/